# Natural history of long-COVID in a nationwide, population cohort study

Claire E. Hastie[1], David J. Lowe [1,2], Andrew McAuley[3,4], Nicholas L. Mills [5,6], Andrew J. Winter [7], Corri Black[8,9], Janet T. Scott[10], Catherine A. O'Donnell [1], David N. Blane [1], Susan Browne[1], Tracy R. Ibbotson[1] & Jill P. Pell [1] ✉

Previous studies on the natural history of long-COVID have been few and selective. Without comparison groups, disease progression cannot be differentiated from symptoms originating from other causes. The Long-COVID in Scotland Study (Long-CISS) is a Scotland-wide, general population cohort of adults who had laboratory-confirmed SARS-CoV-2 infection matched to PCR-negative adults. Serial, self-completed, online questionnaires collected information on pre-existing health conditions and current health six, 12 and 18 months after index test. Of those with previous symptomatic infection, 35% reported persistent incomplete/no recovery, 12% improvement and 12% deterioration. At six and 12 months, one or more symptom was reported by 71.5% and 70.7% respectively of those previously infected, compared with 53.5% and 56.5% of those never infected. Altered taste, smell and confusion improved over time compared to the never infected group and adjusted for confounders. Conversely, late onset dry and productive cough, and hearing problems were more likely following SARS-CoV-2 infection.

Understanding the scale and natural history of long-COVID is essential to planning health and social care. The majority of studies report the prevalence of long-COVID at a single timepoint post-infection[1–7], with some adjusting for pre-existing symptoms[8,9]. Less is known about changes in long-COVID over time. Studies with serial outcome measurements have been restricted to selected groups (e.g. hospitalised patients[10–13], older patients[14], or veterans with break-through infections[15]) or specific (e.g. mental health) outcomes[16], or have lacked a comparison group making it difficult to distinguish persistent or late-onset symptoms of long-COVID from symptoms that would have occurred anyway in the absence of SARS-CoV-2 infection[11–13,17,18].

Hospital cohorts have variously reported no change[10,11,13], improvement[10,11], and deterioration over time[10,14]. In one cohort of 807 people, there was no change in the proportion reporting full recovery

between 5 months and 1 year post discharge[13]. A study of 61 subjects reported no change in quality of life, but improvement in 6-min walking test distance[11]. A meta-analysis of seven cohort studies included 2883 people with repeat measures, in whom the prevalence of depression declined over follow-up and was not significantly different to the comparison group beyond 2 months[16].

Some studies have highlighted the possibility of late-onset sequelae. In an ambidirectional cohort study, conducted on 1276 patients, the proportion reporting at least one symptom decreased from 68% at 6-month follow-up to 49% at 12 months[10]. However, both breathlessness (26% to 30%) and anxiety/depression (23% to 26%) increased over time. Three- to six-month follow-up of veterans with break-through SARS-CoV-2 infections (infections despite vaccination) revealed increased risk of new (HR 1.13, 95% CI 1.07–1.20) as well as

[1]School of Health and Wellbeing, University of Glasgow G12 8TB, Glasgow, UK. [2]Emergency Department, Queen Elizabeth University Hospital, Glasgow G52 4TF, UK. [3]Public Health Scotland, Meridian Court, Glasgow G2 6QQ, UK. [4]School of Health and Life Sciences, Glasgow Caledonian University, Glasgow G4 0BA, UK. [5]BHF Centre for Cardiovascular Science, University of Edinburgh, Edinburgh EH16 4SU, UK. [6]Usher Institute, University of Edinburgh, Edinburgh EH16 4UX, UK. [7]Sandyford Sexual Health Services, NHS Greater Glasgow and Clyde, Glasgow G3 7NB, UK. [8]Aberdeen Centre for Health Data Science, University of Aberdeen AB25 2ZD, Aberdeen, UK. [9]Public Health Directorate, NHS Grampian, AB15 6RE Aberdeen, UK. [10]MRC-University of Glasgow Centre for Virus Research, University of Glasgow, Glasgow G61 1QH, UK. ✉e-mail: Jill.pell@glasgow.ac.uk

persistent (HR 1.90, 95% CI 1.77–2.04) symptoms[15]. In a retrospective cohort study using linked electronic health records, 37% of people had at least one of nine long-COVID features 3–6 months after SARS-CoV-2 infection[19]. Of these, 40% had not had these features in the first 3 months of follow-up. A hospital cohort of 1438 patients, 60 years of age or older, reported increased risk of progressive and late onset cognitive decline at 12-months follow-up among severe cases[14]. In an online survey of 3762 participants who had previous suspected or confirmed SARS-CoV-2 infection, 86% reported re-occurrence of symptoms over time[17].

Therefore, while long-COVID may be a stable condition in some, existing evidence suggests that others may experience recovery, relapse, or progression. We use serial questionnaire data from the long-COVID in Scotland Study (Long-CISS)[20] to investigate the natural history of long-COVID in an unselected, general population cohort with laboratory-confirmed SARS-CoV-2 infection compared with symptoms in an age-, sex-, and socioeconomically-matched group of people who have never been infected.

## Results

Of the 4,049,590 questionnaires sent out, 345,673 (9%) were completed by 288,173 unique individuals, of whom 257,341 (89%) consented to record linkage, required to obtain their test result (Fig. 1). Following linkage, 53,530 were excluded because they reported a previous positive test that was not recorded on the database, 5687 because they had asymptomatic infections, and 37,343 because they were recruited beyond 6 months follow-up. Of the remaining 160,781 individuals, 80,332 (50%) had previous symptomatic, laboratory-confirmed SARS-CoV-2 infection and 80,449 (50%) had never had a positive test for SARS-CoV2 infection. Of the 80,332 people who had previous symptomatic infections, 12,947 have so far completed questionnaires at both 6- and 12-month follow-up and 4196 have completed questionnaires at both 6- and 18-month follow-up. The corresponding figures for the 80,449 individuals never infected were 11,026 and 1711, respectively. The sample size was 23,973 (12,947 symptomatic infected and 11,026 never infected) for the primary analysis of 6- and 12-month follow-up. The index test

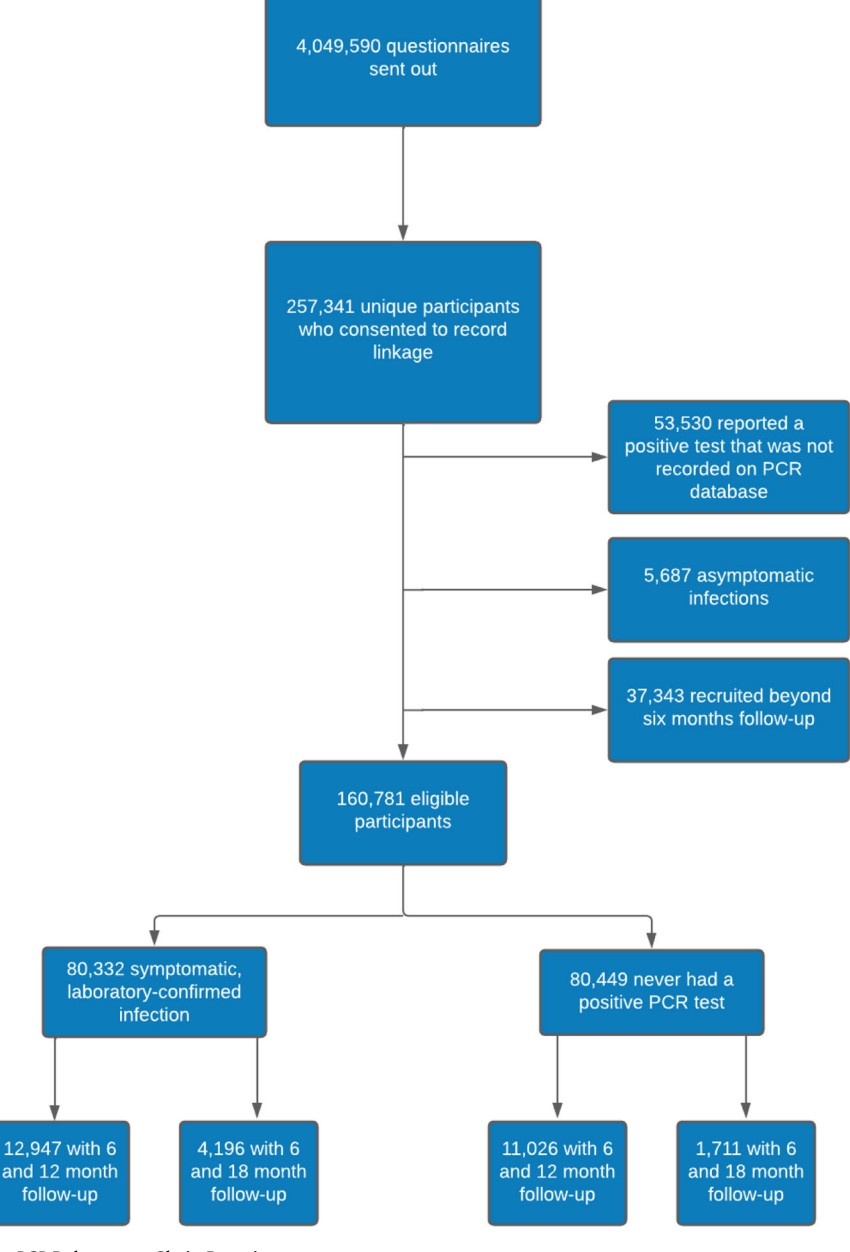

**Fig. 1 | Participant flow diagram.** *PCR* Polymerase Chain Reaction.

**Table 1 | Trajectories of recovery status following symptomatic SARS-CoV-2 infection**

| Recovery status | | 6 and 12 months N = 12,947 | | 6 and 18 months N = 4196 | |
|---|---|---|---|---|---|
| | | N | % | N | % |
| Constant | Overall | 9839 | 76% | 2998 | 71% |
| | Full to full | 5368 | 55% | 1504 | 50% |
| | Partial to partial | 4012 | 41% | 1324 | 44% |
| | No to no | 459 | 4.7% | 170 | 5.7% |
| Deteriorated | Overall | 1497 | 12% | 587 | 14% |
| | Full to partial | 996 | 67% | 386 | 66% |
| | Full to no | 43 | 2.9% | 12 | 2.0 |
| | Partial to no | 458 | 31% | 189 | 32% |
| Improved | Overall | 1611 | 12% | 611 | 15% |
| | No to partial | 404 | 25% | Not disclosed | Not disclosed |
| | No to full | 28 | 1.7% | Not disclosed | Not disclosed |
| | Partial to full | 1179 | 73% | 421 | 69% |

Value is not disclosed if <10, or the number in a group <10 can be calculated from other values.

dates of included participants ranged from 20 April 2020 to 30 November 2021.

## Changes in recovery status

Six months following SARS-CoV-2 infection, 6407 (49.5%) people reported being fully recovered, 5649 (43.6%) partially and 891 (6.9%) not recovered. At 12-month follow-up, the figures were 6575 (50.8%), 5412 (41.8%) and 960 (7.4%), respectively ($\chi^2$ trend, $p = 0.323$). Forty-one percent of people reported full recovery at both 6- and 12-month follow-up, 35% reported persistent incomplete/no recovery, 12% reported improvement and 12% deterioration (Table 1). Between 6 and 18 months, the figures were 36%, 36%, 14% and 15%, respectively.

Of those who felt partially recovered at 6 months, 1179/5649 (21%) had improved by 12 months and 421/1934 (22%) by 18 months and, of the 891 people not recovered at 6 months, 404 (45%) had some degree of improvement by 12 months, and 28 (3%) had fully recovered. Of those who felt partially recovered at 6 months, 458/5649 (8%) reported deterioration at 12 months and 189/1934 (10%) at 18 months. In addition, of 6407 who reported being fully recovered at 6 months, 1039 (16%) reported deterioration by 12 months.

Depression prior to SARS-CoV-2 infection and socioeconomic deprivation were more common among people who reported deterioration in recovery status between 6 and 12 months (Table 2). Similar patterns were observed comparing 6- and 18-month follow-up (Supplementary Table 1). Among those not fully recovered at 6 months, improvement at 12 months was less likely among older people and those with depression prior to COVID-19 and more likely among the most affluent, after adjusting for potential confounders (Table 3; unadjusted odds ratios in Supplementary Table 2). Among those who reported full or partial recovery at 6 months, deterioration at 12 months was less likely among older people and the most affluent and more likely among people with prior depression (Table 3). The associations were not statistically significant comparing 6- and 18-month follow-up (Supplementary Table 3; unadjusted odds ratios in Supplementary Table 4).

## Changes in symptoms

The percentage who reported at least one of the 26 symptoms did not change between 6- and 12-month (72% versus 71%, respectively), and 6-

and 18-month (73% versus 74%, respectively) follow-up, among people with previous symptomatic SARS-CoV-2 infection but increased significantly among those never infected (54% versus 57%, and 52% versus 55%, respectively; Table 4). Previously symptomatic participants had a higher prevalence of new and persistent symptoms than those never infected, at both 12- and 18-month follow-up compared with 6-month follow-up (Supplementary Fig. 2).

The prevalence of confusion and altered taste and smell decreased significantly between 6 and 12 months after SARS-CoV-2 infection contrasting with no significant change in confusion and altered smell, and an increase in altered taste, among those never infected (Table 4). The reductions were significant compared to those never infected after adjusting for potential confounders (Table 5; unadjusted odds ratios in Supplementary Table 5).

Reduced prevalence of altered taste/smell and confusion was specific to those who reported an improvement in their recovery status following SARS-CoV-2 infection (Supplementary Table 6). The prevalence of confusion 6 months following symptomatic SARS-CoV-2 infection was significantly higher among those with a history of depression or anxiety than those without (1090/5839 (18.7%) versus 780/7108 (11.0%); $p < 0.001$) and improvement in confusion between 6- and 12-months was less likely among people with pre-existing depression or anxiety (Table 5).

People with previous symptomatic SARS-CoV-2 infection reported significant increases in the prevalence of both dry and productive cough between 6- and 12-month follow-up (Table 4). However, these symptoms were also reported more frequently over time in the never infected group. The increased prevalence of both dry and productive cough remained significantly higher among those previously infected than those never infected, after adjusting for confounders (Table 5). The factors associated with increased prevalence of dry cough were younger age, more pre-existing long-term conditions, and specifically pre-existing depression/anxiety (secondary effect estimates; Table 5). Increased prevalence of productive cough was associated with male sex and pre-existing respiratory disease (secondary effect estimates; Table 5). Following SARS-CoV-2 infection, late onset cough was specific to those who reported deterioration in their recovery status (Supplementary Table 6).

Increases in the prevalence of hearing problems between 6- and 12-month follow-up were reported by both those with previous symptomatic SARS-CoV-2 infection and those never infected (Table 4). After adjustment for confounders, the increased prevalence of hearing problems was significantly higher among those previously infected than those never infected (Table 5). Other factors associated with late onset hearing problems were socioeconomic deprivation, SARS-CoV-2 infection severity, and more pre-existing long-term conditions and specifically depression/anxiety (secondary effect estimates; Table 5).

Between 6- and 18-months follow-up, increased prevalence of dry cough, productive cough and hearing problems were all significant compared to those never infected after adjusting for potential confounders (Supplementary Table 7; unadjusted odds ratios in Supplementary Table 8).

## Changes in quality of life

Following symptomatic SARS-CoV-2 infection, median EQ-5D VAS score decreased slightly from 75 (IQR 55–86) at 6 months to 74 (IQR 53–85) at 12 months ($p < 0.001$). However, it also fell among those never infected, from 80 (IQR 64–90) to 77 (IQR 61–90) ($p < 0.001$). In the fully adjusted Poisson regression model, symptomatic infection was associated with a larger fall in EQ-5D VAS score compared with those never infected (IRR 0.98, 95% CI 0.98–0.98).

## Discussion

This study reports the trajectory of long-COVID in the general population compared to contemporaneous changes in symptoms and

**Table 2 | Characteristics of participants by infection status and recovery status trajectory between 6 and 12 months**

| | Never infected N = 11,026 | Previous SARS-CoV-2 infection | | | |
| --- | --- | --- | --- | --- | --- |
| | | Constant recovery status N = 9839 | Deteriorated recovery status N = 1497 | Improved recovery status N = 1611 | P value* |
| | Median (IQR) | Median (IQR) | Median (IQR) | Median (IQR) | |
| Age (years) | 52 (39–61) | 51 (38–60) | 50 (37–59) | 50 (37–59) | 0.117 |
| **Sex** | N (%) | N (%) | N (%) | N (%) | |
| Female | 6441 (58.4) | 6259 (63.6) | 1015 (67.8) | 1123 (69.7) | <0.001 |
| Male | 4585 (41.6) | 3580 (36.4) | 482 (32.2) | 488 (30.3) | |
| **SIMD** | | | | | |
| 1 (most deprived) | 2257 (20.5) | 2040 (20.7) | 366 (24.5) | 332 (20.6) | 0.001 |
| 2 | 2248 (20.4) | 2016 (20.5) | 333 (22.2) | 347 (21.5) | |
| 3 | 2127 (19.3) | 1822 (18.5) | 251 (16.8) | 316 (19.6) | |
| 4 | 2164 (19.6) | 1921 (19.5) | 298 (19.9) | 315 (19.6) | |
| 5 (least deprived) | 2230 (20.2) | 2040 (20.7) | 249 (16.6) | 301 (18.7) | |
| **Ethnic group** | | | | | |
| White | 10,200 (92.5) | 9343 (95.0) | 1424 (95.1) | 1537 (95.4) | 0.694 |
| South Asian | 141 (1.28) | 110 (1.12) | 20 (1.34) | Not disclosed | |
| Black | 48 (0.44) | 32 (0.33) | Not disclosed | Not disclosed | |
| Other | 176 (1.60) | 110 (1.12) | Not disclosed | 19 (1.18) | |
| Missing | 461 (4.18) | 244 (2.48) | 35 (2.34) | 38 (2.36) | |
| **Number of pre-existing health conditions** | | | | | |
| 0 | 6870 (62.3) | 6619 (67.3) | 969 (64.5) | 1028 (63.8) | 0.002 |
| 1 | 1602 (14.5) | 1427 (14.5) | 215 (14.4) | 250 (15.5) | |
| 2–3 | 1938 (17.6) | 1442 (14.7) | 231 (15.4) | 260 (16.1) | |
| ≥4 | 616 (5.59) | 351 (3.57) | 82 (5.48) | 73 (4.53) | |
| **Pre-existing health conditions** | | | | | |
| Arthritis | 1006 (9.12) | 712 (7.24) | 146 (9.75) | 154 (9.56) | <0.001 |
| Asthma, bronchitis, COPD | 2740 (24.9) | 2217 (22.5) | 379 (25.3) | 409 (25.4) | 0.005 |
| Cancer | 269 (2.44) | 133 (1.35) | 21 (1.40) | 24 (1.49) | 0.903 |
| CHD | 514 (4.66) | 358 (3.64) | 62 (4.14) | 66 (4.10) | 0.470 |
| Cystic fibrosis | Not disclosed | Not disclosed | Not disclosed | Not disclosed | 0.570 |
| Deep vein thrombosis | 57 (0.52) | 37 (0.38) | 10 (0.67) | Not disclosed | 0.091 |
| Depression/anxiety | 5322 (48.3) | 4272 (43.4) | 774 (51.7) | 793 (49.2) | <0.001 |
| Diabetes | 743 (6.74) | 535 (5.44) | 92 (6.15) | 93 (5.77) | 0.497 |
| High blood pressure | 1399 (12.7) | 1090 (11.1) | 190 (12.7) | 196 (12.2) | 0.110 |
| HIV | 20 (0.18) | Not disclosed | Not disclosed | Not disclosed | 0.325 |
| Home oxygen | Not disclosed | Not disclosed | Not disclosed | Not disclosed | 0.018 |
| Kidney disease | 88 (0.80) | 72 (0.73) | 12 (0.80) | 15 (0.93) | 0.685 |
| Liver disease | 81 (0.73) | 33 (0.34) | 10 (0.67) | Not disclosed | 0.146 |
| Neurological condition | 342 (3.10) | 229 (2.33) | 35 (2.34) | 38 (2.36) | 0.997 |
| Overweight | 1271 (11.5) | 978 (9.94) | 161 (10.8) | 188 (11.7) | 0.083 |
| Obese | 509 (4.62) | 336 (3.41) | 52 (3.47) | 59 (3.66) | 0.880 |
| Pulmonary embolism | 56 (0.51) | 36 (0.37) | Not disclosed | Not disclosed | 0.138 |
| Pulmonary fibrosis | 18 (0.16) | Not disclosed | Not disclosed | Not disclosed | 0.189 |
| Stroke | 125 (1.13) | 92 (0.94) | 17 (1.14) | 15 (0.93) | 0.754 |
| **Vaccinated** | | | | | |
| No | 9491 (86.1) | 6423 (65.3) | 1012 (67.6) | 1129 (70.1) | 0.002 |
| 1 dose | 536 (4.86) | 606 (6.16) | 97 (6.48) | 90 (5.59) | |
| ≥2 doses | 999 (9.06) | 2810 (28.6) | 388 (25.9) | 392 (24.3) | |
| **Variant period** | | | | | |
| Pre VOC | 3840 (34.8) | 2738 (27.8) | 407 (27.2) | 450 (27.9) | 0.004 |
| No dominant (1) | 4824 (43.8) | 2801 (28.5) | 484 (32.3) | 511 (31.7) | |
| Alpha | 1013 (9.19) | 495 (5.03) | 72 (4.81) | 93 (5.77) | |

**Table 2 (continued) | Characteristics of participants by infection status and recovery status trajectory between 6 and 12 months**

| | Never infected N = 11,026 | Previous SARS-CoV-2 infection | | | |
| --- | --- | --- | --- | --- | --- |
| | | Constant recovery status N = 9839 | Deteriorated recovery status N = 1497 | Improved recovery status N = 1611 | P value* |
| No dominant (2) | 230 (2.09) | 245 (2.49) | Not disclosed | Not disclosed | |
| Delta | 1102 (9.99) | 3528 (35.9) | 496 (33.1) | 530 (32.9) | |
| No dominant (3) | 17 (0.15) | 32 (0.33) | Not disclosed | Not disclosed | |

Value is not disclosed if <10, or the number in a group <10 can be calculated from other values.

*IQR* inter-quartile range, *N* number, *SIMD* Scottish Index of Multiple Deprivation, *COPD* chronic obstructive pulmonary disease, *CHD* coronary heart disease, *HIV* human immunodeficiency virus, *VOC* variant of concern.

*Comparison of the three trajectories among those with previous SARS-CoV-2 infection. Kruskal Wallis test for continuous variables, Chi$^2$ test for categorical variables. All statistical tests are two-sided.

quality of life in a comparison group that had never been infected. Beyond 6 months following SARS-CoV-2 infection, there was no significant overall change in either self-reported recovery status or the percentage of people reporting at least one symptom known to be associated with previous SARS-CoV-2 infection[20]. However, 12% of people reported improvements in their recovery status, and 12% reported deterioration. These different trajectories were driven by different symptoms. In some people, altered taste, smell and confusion ('brain fog') resolved over time whereas others reported late onset dry or productive cough and hearing problems. These changes were not explained by underlying trends or confounding. Our findings demonstrate the importance of exploring individual symptoms rather than only grouping them together as a composite outcome.

Our analyses of serial outcomes corroborated our previous finding of late onset cough[20], and identified a new finding of late onset hearing problems. Respiratory impairment following COVID-19 is well-recognised. A meta-analysis of 15 studies that followed-up 3066 patients hospitalised for SARS-CoV-2 infection reported that 56% had residual lung CT abnormalities and 44% had abnormal lung function tests: 35% impaired diffusion, 16% restrictive impairment and 8% obstructive impairment[21]. Systematic reviews had demonstrated that sudden sensorineural hearing loss occurring during acute SARS-CoV-2 infection can persist[22]. Proposed mechanisms include a direct effect of viral invasion via ACE2 receptors located in the ear[23], indirect effects via hypoxia, immune-mediated damage or coagulative disorders[24], and ototoxic medications used to treat COVID-19[22]. The prevalence of several common long-COVID symptoms, for example fatigue, muscle aches/weakness, headache, and anxiety/depression, remained stable within previously infected individuals over time. This has implications for individuals living with long-COVID and for clinical practice.

Socioeconomic deprivation and depression are known to be associated with development of long-COVID[4,19,20]. Our findings elaborate by showing both to also be associated with reduced risk of improvement over time and increased risk of deterioration. Biological mechanisms may partly explain these observations. The relationship between depression and inflammation is bidirectional[25–27]. Novel immune therapeutic targets are being investigated for the treatment of depression[28], and there is a well-documented link between acute or chronic psychological stress and immune markers[29]. The stress resulting from socioeconomic deprivation has been linked to changes in immune response and wider detrimental health effects[30,31]. However, in this study, socioeconomic differences in self-reported recovery status were not corroborated by different changes in specific symptoms over time, other than a weak association with hearing problems. Therefore, it is also plausible that more deprived groups have less capacity to adapt their lives to ongoing health problems or poorer access to support.

In our study, 70.7% of previously infected people who provided 12-month follow-up data had at least one symptom at 12 months following SARS-CoV-2 infection. Estimates from previous longitudinal studies ranged from 28%[12] to 77%[18]. Our 12-month prevalence rates of specific symptoms were comparable to those reported in a random-effects meta-analysis of 18 studies[2]: fatigue/weakness (28% versus 28%), dyspnoea (24% versus 18%), arthromyalgia (24% versus 26%), depression (21% versus 23%), and concentration difficulties or confusion (15% versus 18%). However, the high prevalence of these symptoms among people never infected reinforces the importance of a comparison group, as does the increase over time in the prevalence of at least one symptom in this group (from 54% to 57% between 6- and 12-month follow-up, and from 52% to 55% between 6- and 18-month follow-up).

The major strengths of our study included national, non-selective coverage, self-reported plus laboratory-confirmed SARS-CoV-2 infection status, and inclusion of a comparison group, never infected over the same period of the pandemic. Other studies have relied on historic controls or controls sampled earlier in the pandemic[16]. Recovery status, ongoing symptoms and quality of life were not subject to recall bias because participants reported them at the time of completing questionnaires. Follow-up to 18 months is longer than previously reported. While every adult in Scotland with a positive SARS-CoV-2 PCR test was invited to take part in the study, participation was voluntary, so response bias is possible.

A study limitation was that differential attrition by exposure could not be assessed within the permissions of the data sharing agreement because the researchers could not identify the individuals who moved from the negative to infected group. Furthermore, it is possible that some individuals in the comparison group had SARS-CoV-2 infection that was not detected by a PCR test. This risk was reduced by excluding from the analyses 53,530 participants who had only negative PCR tests recorded but who reported that they had had SARS-CoV-2 infection. However, classification error due to undiagnosed, asymptomatic infection remains. A further limitation, associated with any observational study, is residual confounding due to unknown or unmeasured confounders.

In conclusion, while long-COVID appeared to be a stable condition in many, both improvement and deterioration occurred in others. Improvements in altered taste, smell and confusion were reassuring. In contrast, the findings of late-onset cough and hearing problems one year following infection, that could not be explained by background trends or confounding, merit further investigation.

## Methods

### Study design and participants

The Long-COVID in Scotland Study (Long-CISS) is an ambidirectional, general population cohort. The National Health Service Scotland notification platform for SARS-CoV-2 PCR results was used to identify eligible participants and invite them via automated SMS text messages. Every adult (>16 years) in Scotland with a positive PCR test from April 2020 was invited along with a comparison group who had had a negative test but never a positive test (hereafter referred to as never

**Table 3 | Binary logistic regression of factors associated with improvement and deterioration in recovery status between 6 and 12 months**

| | | Improvement (excluding those fully recovered at 6 months) | | Deterioration (excluding those with no recovery at 6 months) | |
| --- | --- | --- | --- | --- | --- |
| | | Referent no change | Referent no change plus deterioration | Referent no change | Referent no change plus improvement |
| | | *N* = 6082 OR (95% CI) | *N* = 6540 OR (95% CI) | *N* = 10,877 OR (95% CI) | *N* = 12,056 OR (95% CI) |
| Age | | 0.99 (0.99,1.00) | 0.99 (0.99,1.00) | 0.99 (0.99,1.00) | 0.99 (0.99,1.00) |
| Sex | Female | 1.00 | 1.00 | 1.00 | 1.00 |
| | Male | 0.99 (0.87,1.13) | 0.98 (0.86,1.12) | 0.89 (0.79,1.01) | 0.91 (0.81,1.03) |
| Ethnic group | White | 1.00 | 1.00 | 1.00 | 1.00 |
| | South Asian | 0.94 (0.45,1.95) | 0.93 (0.45,1.91) | 1.23 (0.76,2.01) | 1.30 (0.80,2.11) |
| | Black | 2.22 (0.82,6.04) | 2.43 (0.89,6.62) | 0.78 (0.27,2.22) | 0.74 (0.26,2.07) |
| | Other | 1.52 (0.86,2.68) | 1.53 (0.87,2.69) | 0.85 (0.49,1.50) | 0.86 (0.49,1.51) |
| | Missing | 0.93 (0.64,1.36) | 0.97 (0.67,1.40) | 0.93 (0.65,1.34) | 0.95 (0.66,1.36) |
| SIMD quintile | 1 (most deprived) | 1.00 | 1.00 | 1.00 | 1.00 |
| | 2 | 1.16 (0.97,1.38) | 1.18 (0.97,1.41) | 0.94 (0.80,1.11) | 0.93 (0.79,1.09) |
| | 3 | 1.28 (1.07,1.54) | 1.33 (1.11,1.59) | 0.79 (0.66,0.94) | 0.77 (0.65,0.92) |
| | 4 | 1.26 (1.06,1.52) | 1.29 (1.08,1.55) | 0.90 (0.76,1.07) | 0.88 (0.75,1.04) |
| | 5 (least deprived) | 1.37 (1.14,1.65) | 1.44 (1.20,1.73) | 0.73 (0.61,0.86) | 0.72 (0.60,0.86) |
| Pre-existing long-term conditions | 0 | 1.00 | 1.00 | 1.00 | 1.00 |
| | 1 | 1.03 (0.87,1.22) | 1.05 (0.89,1.25) | 0.98 (0.83,1.17) | 0.97 (0.82,1.14) |
| | 2–3 | 0.92 (0.77,1.10) | 0.94 (0.78,1.12) | 0.99 (0.83,1.18) | 0.98 (0.82,1.17) |
| | ≥4 | 0.98 (0.71,1.36) | 0.94 (0.68,1.30) | 1.40 (1.03,1.89) | 1.40 (1.04,1.89) |
| Asthma/bronchitis/ COPD | No | 1.00 | 1.00 | 1.00 | 1.00 |
| | Yes | 0.94 (0.82,1.09) | 0.93 (0.80,1.06) | 1.10 (0.95,1.26) | 1.09 (0.95,1.25) |
| CHD | No | 1.00 | 1.00 | 1.00 | 1.00 |
| | Yes | 1.17 (0.86,1.61) | 1.16 (0.85,1.58) | 1.07 (0.79,1.44) | 1.06 (0.79,1.43) |
| Depression/anxiety | No | 1.00 | 1.00 | 1.00 | 1.00 |
| | Yes | 0.81 (0.71,0.91) | 0.78 (0.69,0.88) | 1.35 (1.20,1.51) | 1.34 (1.19,1.50) |
| Diabetes | No | 1.00 | 1.00 | 1.00 | 1.00 |
| | Yes | 1.03 (0.79,1.36) | 1.04 (0.79,1.36) | 1.05 (0.81,1.36) | 1.03 (0.80,1.33) |
| Variant period | preVOC | 1.00 | 1.00 | 1.00 | 1.00 |
| | No dominant (1) | 1.18 (1.01,1.37) | 1.15 (0.99,1.33) | 1.17 (1.01,1.35) | 1.15 (0.99,1.32) |
| | Alpha | 1.27 (0.97,1.65) | 1.29 (0.99,1.68) | 0.96 (0.73,1.26) | 0.93 (0.71,1.22) |
| | No dominant (2) | 0.77 (0.48,1.25) | 0.75 (0.47,1.21) | 0.94 (0.63,1.39) | 0.95 (0.64,1.40) |
| | Delta | 1.47 (1.13,1.92) | 1.46 (1.13,1.90) | 0.96 (0.74,1.25) | 0.92 (0.71,1.19) |
| | No dominant (3) | 0.63 (0.18,2.21) | 0.65 (0.19,2.26) | 0.45 (0.10,1.91) | 0.44 (0.10,1.89) |
| Vaccinated | No | 1.00 | 1.00 | 1.00 | 1.00 |
| | 1 dose | 0.79 (0.59,1.07) | 0.80 (0.59,1.07) | 1.08 (0.82,1.42) | 1.12 (0.86,1.47) |
| | ≥2 doses | 0.79 (0.60,1.04) | 0.78 (0.59,1.03) | 0.99 (0.76,1.29) | 1.05 (0.81,1.37) |
| Infection severity | Not hospitalised | 1.00 | 1.00 | 1.00 | 1.00 |
| | Hospitalised | 0.75 (0.59,0.95) | 0.74 (0.58,0.94) | 1.09 (0.84,1.43) | 1.08 (0.83,1.40) |

Odds ratios are adjusted (unadjusted odds ratios provided in Supplementary Table 2).
Previously infected individuals who provided 6- and 12-month follow-up. The sample size in each column is composed as follows:
Improved referent to no change: 12,947 previously infected with follow-up at 6 and 12 months excluding 6407 people who were already fully recovered at 6 months follow-up (so could not improve) and excluding people whose recovery status deteriorated (458 after deducting those already excluded) = 6082.
Improved referent to no change or deterioration: 12,947 previously infected with follow-up at 6 and 12 months excluding 6407 people who were already fully recovered at 6 months follow-up (so could not improve) = 6540. Other people who deteriorated are included in the referent category so are not excluded.
Deterioration referent to no change: 12,947 previously infected with follow-up at 6 and 12 months excluding 891 people who reported no recovery at 6 months (so could not deteriorate) and excluding people whose recovery status improved (1179 after deducting those already excluded) = 10,877.
Deterioration referent to no change or improvement: 12,947 previously infected with follow-up at 6 and 12 months excluding 891 people who reported no recovery at 6 months (so could not deteriorate) = 12,056. Other people who improved are included in the referent category so are not excluded.
*OR* odds ratio, *CI* confidence interval, *SIMD* Scottish Index of Multiple Deprivation, *COPD* chronic obstructive pulmonary disease, *CHD* coronary heart disease, *VOC* variant of concern. All statistical tests are two-sided.

infected), matched by age, sex, deprivation quintile, and time period of test[20]. People in the latter group were reallocated to the infected group if, and when, they had a positive test. The study commenced in May 2021 and recruited both retrospectively and prospectively based on existing and new test results, respectively. Participants provided electronic consent and study approval was obtained from the West of Scotland Research Ethics Committee (ref. 21/WS/0020) and Public Benefit and Privacy Panel (ref. 2021-0180). The ethics committee deemed that those under 18 (and over 16) years could provide informed consent.

**Table 4 | Prevalence of symptoms reported at 6 versus 12 months follow-up, and 6 versus 18 months follow-up, among people who had symptomatic SARS-CoV-2 infection and those never infected**

| | Symptomatic | | | | | | Never infected | | | | | |
|---|---|---|---|---|---|---|---|---|---|---|---|---|
| | 6 and 12 months N=12,947 | | | 6 and 18 months N=4196 | | | 6 and 12 months N=11,026 | | | 6 and 18 months N=1711 | | |
| | 6 months N (%) | 12 months N (%) | p-value* | 6 months N (%) | 18 months N (%) | p-value* | 6 months N (%) | 12 months N (%) | p-value* | 6 months N (%) | 18 months N (%) | p-value* |
| **Sensory** | | | | | | | | | | | | |
| Altered taste | 1453 (11.2) | 1125 (8.69) | <0.001 | 421 (10.0) | 322 (7.67) | <0.001 | 162 (1.47) | 200 (1.81) | 0.035 | 16 (0.94) | 31 (1.81) | 0.024 |
| Altered smell | 1679 (13.0) | 1301 (10.1) | <0.001 | 483 (11.5) | 339 (8.08) | <0.001 | 116 (1.05) | 145 (1.32) | 0.062 | 17 (0.99) | 21 (1.23) | 0.618 |
| Problems hearing | 761 (5.88) | 867 (6.70) | 0.001 | 244 (5.82) | 303 (7.22) | 0.002 | 373 (3.38) | 433 (3.93) | 0.013 | 57 (3.33) | 75 (4.38) | 0.082 |
| Problems with eyesight | 974 (7.52) | 986 (7.62) | 0.751 | 333 (7.94) | 356 (8.48) | 0.283 | 441 (4.00) | 467 (4.24) | 0.322 | 82 (4.79) | 84 (4.91) | 0.925 |
| Pins and needles | 1468 (11.3) | 1530 (11.8) | 0.129 | 539 (12.9) | 579 (13.8) | 0.109 | 739 (6.70) | 729 (6.61) | 0.775 | 128 (7.48) | 128 (7.48) | 1.000 |
| **Cardiorespiratory** | | | | | | | | | | | | |
| Chest pain | 920 (7.11) | 926 (7.15) | 0.877 | 318 (7.58) | 327 (7.79) | 0.691 | 289 (2.62) | 393 (3.56) | <0.001 | 63 (3.68) | 66 (3.86) | 0.838 |
| Palpitations | 1197 (9.25) | 1215 (9.38) | 0.632 | 444 (10.6) | 460 (11.0) | 0.503 | 429 (3.89) | 443 (4.02) | 0.588 | 75 (4.38) | 73 (4.27) | 0.920 |
| Breathlessness | 2879 (22.2) | 2830 (21.9) | 0.309 | 1082 (25.8) | 1058 (25.2) | 0.431 | 863 (7.83) | 1076 (9.76) | <0.001 | 172 (10.1) | 188 (11.0) | 0.277 |
| Dry cough | 1559 (12.0) | 1946 (15.0) | <0.001 | 484 (11.5) | 713 (17.0) | <0.001 | 761 (6.90) | 1127 (10.2) | <0.001 | 119 (6.95) | 187 (10.9) | <0.001 |
| Cough with phlegm | 1408 (10.9) | 1819 (14.1) | <0.001 | 432 (10.3) | 628 (15.0) | <0.001 | 938 (8.51) | 1382 (12.5) | <0.001 | 146 (8.53) | 200 (11.7) | <0.001 |
| **Gastrointestinal** | | | | | | | | | | | | |
| Poor appetite | 747 (5.77) | 776 (5.99) | 0.375 | 246 (5.86) | 246 (5.86) | 1.000 | 480 (4.35) | 527 (4.78) | 0.088 | 82 (4.79) | 79 (4.62) | 0.853 |
| Abdominal pain | 930 (7.18) | 981 (7.58) | 0.149 | 324 (7.72) | 342 (8.15) | 0.413 | 701 (6.36) | 745 (6.76) | 0.173 | 117 (6.84) | 119 (6.95) | 0.936 |
| Sickness/vomiting | 812 (6.27) | 832 (6.43) | 0.569 | 266 (6.34) | 324 (7.72) | 0.004 | 551 (5.00) | 578 (5.24) | 0.381 | 96 (5.61) | 104 (6.08) | 0.570 |
| Diarrhea | 1092 (8.43) | 1138 (8.79) | 0.240 | 385 (9.18) | 424 (10.1) | 0.094 | 812 (7.36) | 815 (7.39) | 0.953 | 117 (6.84) | 119 (6.95) | 0.935 |
| Constipation | 658 (5.08) | 649 (5.01) | 0.788 | 238 (5.67) | 225 (5.36) | 0.511 | 434 (3.94) | 496 (4.50) | 0.017 | 79 (4.62) | 80 (4.68) | 1.000 |
| **Musculoskeletal** | | | | | | | | | | | | |
| Muscle aches/weakness | 3387 (26.2) | 3446 (26.6) | 0.271 | 1231 (29.3) | 1269 (30.2) | 0.236 | 1723 (15.6) | 1881 (17.1) | <0.001 | 299 (17.5) | 307 (17.9) | 0.702 |
| Joint pain | 2798 (21.6) | 2831 (21.9) | 0.516 | 1020 (24.3) | 1092 (26.0) | 0.018 | 1784 (16.2) | 1869 (17.0) | 0.054 | 293 (17.1) | 336 (19.6) | 0.019 |
| **Neurological/mental health** | | | | | | | | | | | | |
| Headache | 3250 (25.1) | 3325 (25.7) | 0.184 | 1141 (27.2) | 1145 (27.3) | 0.927 | 2131 (19.3) | 2423 (22.0) | <0.001 | 307 (17.9) | 337 (19.7) | 0.144 |
| Anxious/depressed | 2588 (20.0) | 2559 (19.8) | 0.568 | 921 (22.0) | 904 (21.5) | 0.569 | 1460 (13.2) | 1566 (14.2) | 0.008 | 232 (13.6) | 248 (14.5) | 0.337 |
| Confusion | 1870 (14.4) | 1757 (13.6) | 0.007 | 723 (17.2) | 637 (15.2) | <0.001 | 534 (4.84) | 532 (4.82) | 0.970 | 101 (5.90) | 95 (5.55) | 0.646 |
| Sleep problems | 3381 (26.1) | 3437 (26.6) | 0.310 | 1149 (27.4) | 1198 (28.6) | 0.131 | 1942 (17.6) | 2130 (19.3) | <0.001 | 300 (17.5) | 293 (17.1) | 0.736 |
| Dizzy/blackouts/fits | 653 (5.04) | 645 (4.98) | 0.812 | 234 (5.58) | 233 (5.55) | 1.000 | 330 (2.99) | 354 (3.21) | 0.308 | 44 (2.57) | 49 (2.86) | 0.649 |
| Balance problems | 901 (6.96) | 980 (7.57) | 0.018 | 317 (7.55) | 356 (8.48) | 0.058 | 412 (3.74) | 465 (4.22) | 0.028 | 79 (4.62) | 95 (5.55) | 0.145 |
| **Non-specific** | | | | | | | | | | | | |
| Tiredness | 6055 (46.8) | 5987 (46.2) | 0.264 | 2110 (50.3) | 2145 (51.1) | 0.325 | 3360 (30.5) | 3541 (32.1) | 0.001 | 505 (29.5) | 534 (31.2) | 0.193 |
| Weight loss | 280 (2.16) | 241 (1.86) | 0.047 | 103 (2.45) | 87 (2.07) | 0.221 | 132 (1.20) | 147 (1.33) | 0.344 | 27 (1.58) | 31 (1.81) | 0.652 |
| Skin rash | 552 (4.26) | 578 (4.46) | 0.366 | 167 (3.98) | 212 (5.05) | 0.006 | 277 (2.51) | 314 (2.85) | 0.091 | 43 (2.51) | 52 (3.04) | 0.343 |
| **At least one symptom now** | 9252 (71.5) | 9148 (70.7) | 0.060 | 3080 (73.4) | 3117 (74.3) | 0.243 | 5,899 (53.5) | 6226 (56.5) | <0.001 | 895 (52.3) | 947 (55.4) | 0.026 |

*Two-sided McNemar's test; N number.

**Table 5 | Binary logistic regression models of the factors associated with symptoms at 12 months adjusted for symptoms at 6 months**

| | | Altered taste N = 1614 | Altered smell N = 1794 | Confusion N = 2404 | Hearing problems N = 22,839 | Dry cough N = 21,653 | Cough with phlegm N = 21,627 |
|---|---|---|---|---|---|---|---|
| | | OR (95% CI) | OR (95% CI) | OR (95% CI) | OR (95% CI) | OR (95% CI) | OR (95% CI) |
| Covid-19 status | Never infected | 1.00 | 1.00 | 1.00 | 1.00 | 1.00 | 1.00 |
| | Symptomatic infection | 0.22 (0.14,0.35) | 0.20 (0.12,0.34) | 0.43 (0.35,0.54) | 1.58 (1.35,1.84) | 1.41 (1.28,1.56) | 1.15 (1.05,1.27) |
| Age (years) | | 0.99 (0.99,1.01) | 1.00 (0.99,1.01) | 1.00 (0.99,1.01) | 1.00 (0.99,1.01) | 0.99 (0.99,0.99) | 0.99 (0.98,0.99) |
| Sex | Female | 1.00 | 1.00 | 1.00 | 1.00 | 1.00 | 1.00 |
| | Male | 0.91 (0.72,1.15) | 0.92 (0.74,1.14) | 0.98 (0.81,1.18) | 0.92 (0.79,1.07) | 0.96 (0.87,1.06) | 1.16 (1.05,1.27) |
| Ethnic group | White | 1.00 | 1.00 | 1.00 | 1.00 | 1.00 | 1.00 |
| | South Asian | 1.35 (0.33,5.62) | 0.33 (0.05,1.98) | 1.13 (0.54,2.37) | 0.23 (0.06,0.93) | 0.99 (0.66,1.51) | 1.03 (0.69,1.55) |
| | Black | – | – | 2.51 (0.48,13.2) | 0.64 (0.16,2.64) | 0.63 (0.25,1.56) | 0.36 (0.11,1.16) |
| | Other | 1.53 (0.52,4.55) | 1.84 (0.73,4.63) | 0.69 (0.33,1.48) | 1.02 (0.54,1.93) | 0.73 (0.47,1.15) | 1.02 (0.69,1.50) |
| | Missing | 1.02 (0.53,1.96) | 1.51 (0.82,2.79) | 1.35 (0.81,2.23) | 0.90 (0.59,1.37) | 0.74 (0.56,0.99) | 0.95 (0.73,1.23) |
| SIMD quintile | 1 (most deprived) | 1.00 | 1.00 | 1.00 | 1.00 | 1.00 | 1.00 |
| | 2 | 0.84 (0.62,1.14) | 0.96 (0.72,1.28) | 0.83 (0.65,1.05) | 0.93 (0.77,1.14) | 1.07 (0.94,1.23) | 1.07 (0.94,1.22) |
| | 3 | 0.82 (0.60,1.12) | 0.99 (0.74,1.34) | 0.93 (0.72,1.20) | 0.82 (0.67,1.02) | 1.02 (0.89,1.18) | 0.92 (0.80,1.06) |
| | 4 | 0.99 (0.72,1.35) | 1.16 (0.86,1.55) | 0.90 (0.69,1.16) | 0.86 (0.70,1.07) | 0.95 (0.83,1.09) | 0.96 (0.84,1.11) |
| | 5 (least deprived) | 1.07 (0.77,1.48) | 1.02 (0.75,1.38) | 1.08 (0.83,1.42) | 0.71 (0.57,0.89) | 0.85 (0.74,0.98) | 0.81 (0.70,0.94) |
| Pre-existing long-term conditions | 0 | 1.00 | 1.00 | 1.00 | 1.00 | 1.00 | 1.00 |
| | 1 | 0.90 (0.66,1.23) | 0.85 (0.62,1.15) | 1.19 (0.93,1.52) | 1.07 (0.86,1.33) | 1.45 (1.27,1.65) | 1.23 (1.07,1.40) |
| | 2–3 | 1.16 (0.83,1.62) | 0.79 (0.58,1.09) | 0.96 (0.76,1.22) | 1.69 (1.39,2.04) | 1.52 (1.32,1.73) | 1.58 (1.38,1.80) |
| | ≥4 | 0.66 (0.37,1.18) | 0.71 (0.40,1.29) | 0.87 (0.60,1.26) | 2.43 (1.80,3.29) | 2.32 (1.87,2.88) | 1.98 (1.58,2.47) |
| Asthma/bronchitis/COPD | No | 1.00 | 1.00 | 1.00 | 1.00 | 1.00 | 1.00 |
| | Yes | 0.92 (0.72,1.18) | 1.05 (0.82,1.34) | 0.91 (0.75,1.11) | 1.09 (0.92,1.28) | 1.10 (0.99,1.23) | 1.60 (1.44,1.78) |
| CHD | No | 1.00 | 1.00 | 1.00 | 1.00 | 1.00 | 1.00 |
| | Yes | 1.12 (0.60,2.11) | 1.77 (0.91,3.48) | 0.86 (0.55,1.35) | 1.34 (1.01,1.79) | 1.17 (0.95,1.45) | 1.06 (0.85,1.32) |
| Depression/anxiety | No | 1.00 | 1.00 | 1.00 | 1.00 | 1.00 | 1.00 |
| | Yes | 0.97 (0.78,1.20) | 1.04 (0.85,1.27) | 0.76 (0.63,0.91) | 1.39 (1.20,1.61) | 1.18 (1.07,1.30) | 1.19 (1.08,1.31) |
| Diabetes | No | 1.00 | 1.00 | 1.00 | 1.00 | 1.00 | 1.00 |
| | Yes | 0.77 (0.46,1.29) | 0.83 (0.49,1.39) | 1.07 (0.75,1.53) | 0.80 (0.61,1.06) | 0.95 (0.79,1.15) | 0.93 (0.77,1.13) |
| Variant period | preVOC | 1.00 | 1.00 | 1.00 | 1.00 | 1.00 | 1.00 |
| | No dominant (1) | 1.30 (0.98,1.71) | 1.02 (0.79,1.33) | 1.16 (0.94,1.42) | 0.89 (0.75,1.05) | 0.88 (0.79,0.98) | 0.79 (0.71,0.88) |
| | Alpha | 1.10 (0.64,1.88) | 1.09 (0.63,1.89) | 0.86 (0.61,1.22) | 0.80 (0.58,1.10) | 0.92 (0.75,1.11) | 0.80 (0.66,0.98) |
| | No dominant (2) | 0.96 (0.49,1.86) | 0.62 (0.33,1.15) | 0.64 (0.34,1.23) | 0.83 (0.48,1.45) | 1.01 (0.73,1.39) | 0.59 (0.41,0.86) |
| | Delta | 1.23 (0.80,1.88) | 0.81 (0.55,1.20) | 1.27 (0.84,1.92) | 0.99 (0.68,1.44) | 0.86 (0.68,1.09) | 0.63 (0.49,0.81) |
| | No dominant (3) | 3.78 (0.73,19.5) | 1.21 (0.32,4.53) | 0.89 (0.05,15.0) | 0.98 (0.22,4.23) | 0.86 (0.33,2.24) | 0.85 (0.35,2.10) |
| Vaccinated | No | 1.00 | 1.00 | 1.00 | 1.00 | 1.00 | 1.00 |
| | 1 dose | 0.78 (0.46,1.32) | 0.89 (0.54,1.46) | 0.98 (0.66,1.48) | 0.96 (0.66,1.40) | 0.95 (0.75,1.20) | 1.01 (0.78,1.30) |
| | ≥2 doses | 0.83 (0.53,1.31) | 1.11 (0.73,1.67) | 1.11 (0.72,1.71) | 1.01 (0.69,1.47) | 1.13 (0.89,1.44) | 1.36 (1.04,1.76) |
| Infection severity | Not hospitalised | 1.00 | 1.00 | 1.00 | 1.00 | 1.00 | 1.00 |
| | Hospitalised | 1.36 (0.79,2.33) | 0.87 (0.52,1.48) | 0.77 (0.56,1.06) | 1.69 (1.23,2.34) | 0.98 (0.74,1.29) | 1.23 (0.94,1.60) |

*OR* odds ratio, *CI* confidence interval, *SIMD* Scottish Index of Multiple Deprivation, *COPD* chronic obstructive pulmonary disease, *CHD* coronary heart disease, *VOC* variant of concern. All statistical tests are two-sided.
Odds ratios are adjusted.

An online questionnaire (Supplementary Fig. 1), self-completed 6, 12 and 18 months after the index test (first positive test or, for comparison group, most recent negative test), collected information on pre-existing health conditions and 26 current symptoms (harmonised with the ISARIC questionnaire)[32] and health-related quality of life using the EuroQoL-5D visual analogue scale (EQ-5D VAS) score. Respondents who had tested positive also self-assessed their current recovery status (fully, partially or not recovered). The questionnaire data were linked retrospectively to the PCR test, vaccination, hospital admissions

(Scottish Morbidity Record; SMR01 and SMR04) and dispensed prescriptions (Prescribing Information System; PIS) databases.

**Exclusion and inclusion criteria**

Only people who completed a 6-month follow-up questionnaire plus at least one subsequent questionnaire (12-month, 18-month or both) were included. We excluded people who had asymptomatic SARS-CoV-2 infection, usually detected during occupational or travel-related screening, and those who reported a positive test not recorded on

the database, as we could not corroborate the accuracy and date of tests performed outside Scotland.

## Definitions

Linkage to the test database provided date and result of the index PCR test plus age, sex and postcode of residence. The latter was used to derive the Scottish Index of Multiple Deprivation (SIMD) from aggregated data on: income, employment, education, health, access to services, crime and housing[33]. Severe infection was defined as hospital admission with an International Classification of Diseases v10 (ICD-10) code U07.1 between 1 day prior to the index test and 2 weeks after. Vaccination status (0, 1 or ≥2 doses) at the time of the index test was obtained via linkage to the vaccination database. SARS-CoV-2 variants were defined as dominant if they accounted for ≥95% of cases genotyped that week (https://sars2.cvr.gla.ac.uk/cog-uk/). Ethnic group was self-reported using the questionnaire. Pre-existing health conditions were ascertained from self-report using the questionnaire, as well as linkage to previous hospitalizations and dispensed prescriptions. Respiratory disease was defined as ICD10 codes J40-J47, J98.2 or J98.3, or bronchodilators, inhaled corticosteroids, cromoglycate, leukotriene or phosphodiesterase type-4 inhibitor (British National Formulary (BNF) 3.1-3.3), or self-report. Coronary heart disease was defined as ICD10 codes I11.0, I13.0, I13.2, I20-I25 (excluding I24.1), I50, T82.2, or Z95.5, or self-report. Depression was defined as ICD10 codes F30-F33, or anti-depressant, hypnotic or anxiolytic use (BNF 4.1;4.3), or self-report. Diabetes was defined as ICD10 codes E10-E14, G590, G632, H280, H360, M142, N083, O240-O243 or self-report[34]. Total number of self-reported health conditions was categorised as 0, 1, 2–3 or ≥4.

The outcomes measured were changes in self-reported recovery status following symptomatic SARS-CoV-2 infection, and changes in 26 individual symptoms and quality of life score compared with those never infected. Improvement in recovery status was defined as change from no recovery to partial/full recovery, or from partial recovery to full recovery. Deterioration was defined as change from full recovery to partial/no recovery or from partial recovery to no recovery.

## Statistical analyses

Participant characteristics were summarized using frequencies/percentages and medians/inter-quartile ranges for categorical and continuous variables and compared using $\chi^2$ and Mann–Whitney U tests, respectively. The analyses were first conducted comparing 6- and 12-month follow-up, then repeated comparing 6 and 18 months. Separate binary logistic regression models were used to determine the factors associated with improvement and deterioration in recovery status over time following symptomatic infection; univariately then adjusted for covariates (age, sex, deprivation quintile, ethnic group, individual and total number of long-term conditions, vaccination status, and dominant variant).

Change in the prevalence of the 26 individual symptoms from 6 to 12 months was compared in those with previous, symptomatic SARS-CoV-2 infection and those never infected using McNemar's tests. Separate binary logistic regression models were run for the presence of each symptom at 12 months. The models were adjusted for whether the person had been infected or never infected, whether the symptom was present at 6 months, as well as the confounders listed above. The analyses were repeated for the change in prevalence of individual symptoms from 6 to 18 months.

Median EQ-5D VAS score was calculated at 6 and 12 months for the infected and never infected groups. A Poisson regression model was run for EQ-5D VAS score at 12 months, adjusting for whether the person had been infected or never infected, the score at 6 months, and the confounders listed above. All analyses were performed using Stata v16. All statistical tests were two-tailed.

## Reporting summary

Further information on research design is available in the Nature Portfolio Reporting Summary linked to this article.

## Data availability

The datasets analysed during the current study are available in the National Services Scotland National Safe Haven, https://www.isdscotland.org/Products-and-Services/eDRIS/Use-of-the-National-Safe-Haven/. This protects the confidentiality of the data and ensures that Information Governance is robust. Applications to access health data in Scotland are submitted to the NHS Scotland Public Benefit and Privacy Panel for Health and Social Care. Information can be found at https://www.informationgovernance.scot.nhs.uk/pbpphsc/.

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

## Acknowledgements
The study was funded by the Chief Scientist Office (ref COV/LTE/20/06) and Public Health Scotland. This research used data assets made available by National Safe Haven as part of the Data and Connectivity National Core Study, led by Health Data Research UK in partnership with the Office for National Statistics and funded by UK Research and Innovation (grant ref MC_PC_20029). We are grateful to Public Health Scotland and e-DRIS for providing and linking secondary data and providing a secure analytical environment, Storm-ID for administering invitations and data collection, the Scottish Government for supporting the study launch, and the University of Glasgow, College of Medical, Veterinary and Life Sciences PPIE (Patient and Public Involvement and Engagement) and COVID-19 PPIE groups for their contributions to study design, recruitment, and interpretation of results.

## Author contributions
J.P.P. had the original concept. J.P.P., D.J.L., A.J.W., C.E.H., C.A.O'D., D.N.B., N.L.M., C.B., J.T.S., T.R.I. and A.Mc.A. obtained funding. C.E.H., J.P.P., D.J.L. and A.Mc.A. obtained approvals. C.E.H. analysed the data. J.P.P., D.J.L., A.J.W., C.E.H., C.A.O'D., D.N.B., N.L.M., C.B., J.T.S., T.R.I., A.Mc.A. and S.B. interpreted the results. J.P.P. and C.E.H. produced the first draft. All authors revised the manuscript and approved the final version.

## Competing interests
The authors declare no competing interests.
