## [Peer Review File · Nature Communications]

Natural history of long-COVID in a nationwide, population cohort studyREVIEWER COMMENTS

Reviewer #1 (Remarks to the Author):

This is a valuable study to help understand how adults with on-going symptoms following SARS-CoV-2 infection and one of the first to look at symptoms at 12 and 18 months in a community based population. However, there are a number of detailed comments below that could strengthen the paper. One overall comment, is that it is difficult to track how participants are followed over time and who is and isn't included in the denominator (and numerator) for each estimate. A clear flow diagram on documenting which participants are including in the analyses is needed. In terms of presentation of the results, recommend reorganizing the results to strength the messages of the paper.

Recommendations:

Table 1- comparing characteristics of participants at each time point (6 month, 12 month, 18 months) by infected and not infected status. This will support whether there was differential drop out of participants by infectious status.

Table 2- look at change in report of symptoms from 6 months at 12 months, and then at 18 months- also by infection status.

Figure looking at persistent, recovered, worsening symptoms from 6 to 12 months, and then from 6 to 18 months, by infectious status. Also consider looking at symptoms from 12 to 18 months. Please see comments for how to approach the multivariate analysis.

Specific comments:

Summary of article:

- The authors state that the not infected group was matched to the infected group. If so, what factors were they matched on? If not, then recommend revising this wording.
- Is there any information on symptoms at time of infection or testing?
- Please explain how "fully, partially and not recovered" compares to "persistent and recovered". Based on the methods it looks like fully and partially recovered is based on self assessment, but then persistent symptoms are based on reporting symptoms at 2 time points. Is that correct? This applies to reporting in the main results also.
- Please see the comments related to prevalence of persistent symptoms in the main results and apply here also.

Methods:

- What was the sample size and response rate?
- Can you provide the survey instrument in an on-line supplement?
- Were pre-existing health conditions based on pre-existing conditions prior to SARS-CoV-2 infection or prior to 6 month survey?
- Was linkage to ambulatory care also included?
- What is the completion rate of linkages?
- If a participant who tested negative later tested positive how was this addressed by the study?
- Were quality of life scores asked for both those infected and not infected?
- Were the quality of life scores reported in the results?
- Did you consider analysis where you looked at symptoms at all three time points- 6, 12 and 18 months?
- In the paragraph describing how the change in prevalence was calculated, it is not clear what was done. Suggest revising to state clearly- e.g. "change in prevalence of symptoms from 6 to 12 months was compared among those infected and never infected.... Next change in prevalence of symptoms between 6 and 18 months was compared.... Finally, trend in symptoms from 6 to 12 to 18 months was compared..."
- Why was poisson regression chosen for this analysis? What other factors were included in the model?
- Recommend that models be stratified by infection status. This will allow a comparison of what factors are related to symptoms at the different time points among those not infected and those infected separately.

Results:

- Strongly recommend adding a flow diagram for the participants, both infected and not infected, including who remains in the study at 12 and 18 months.
- Were there any differences in loss to follow-up by infection status of demographic characteristics? If there were differences, how was this addressed in the analysis and how did this likely impact the results?
- This manuscript would benefit from a graph that is based on 18 month follow-up for population (denominator) with a stacked bar with % no symptoms at 18 months, % new symptoms from 6 months, % persistent symptoms from 6 months; could do a similar figure for 12 months. Figures could be shown for symptoms overall and by type and could also produce figures by demographics.
- Recommend a table that reports the characteristics of participants SARS-CoV-2 + at 6 months, then looks at who also completed the survey at 12 m and then 18 m to look at loss of follow-up and how this may have changed across characteristics over time. If there are changes, this should be address in analysis or mentioned in limitations.
- Table 1. Are the report of resolved symptoms in table 1 based on response that specific survey question and then compared to whether or not participants reported symptoms at both time points? How does this compared to the not infected group.
- Table 2. Recommend moving this to the first table and including report of symptoms in the never infected group have survey questions at multiple time points.
- Table 3. Please clarify how the sample in table 3 aligns with what is reported in table 1 and 2.
- The prevalence of symptoms in the never infected group is 53% compared to the infected group this is a difference of 18% which is what has been reported in multiple other studies. given the strength of this study is the never infected group, this comparison should be reported in the results.
- Did you also look at report of 2 or more symptoms?
- Table 5. This model doesn't make sense to me- what exactly is the outcome? symptoms at 12 months among participants reporting the symptom at 6 months? if so then one would say, among participants who had altered taste at 6 months, those who had infection were less likely to have altered taste at 12month compared to those not infected (after adjusting for xxxx) this does make sense with table 4- but looking at table 4 it seems like it may be driven by small numbers in the never infected group
- why not stratify by infection status -then report out factors associated with persisted symptoms if infected, factors associated with persistent symptoms in general?
- Supplemental Table 4 is recovery defined at 12 months? I am unclear based on this table. is change in taste at 6 months change from baseline?

Discussion:

- Paragraph 1 in the discussion describing the trajectories I would frame this as each symptom as a different trajectory and that because of this it is important to look at individual symptoms and not group together.
- In reporting out the prevalence of symptoms at 12 and 18 months, it is often not clear to who is included in the denominator is for the estimate. For example, is it among people who had a symptom at 6 months, meaning that 70% of symptoms persisted or is it among all people who were sars-cov-2 positive? Please clarify this, both in the results and discussion.
- It is also important to compare the estimated prevalence among the infected group with the not infected group. The prevalence of symptoms in the not infected group is also high.
- What does the difference over time in symptoms look like in the not infected group? This is an important point to include in the discussion and to help put the results into context.

Reviewer #2 (Remarks to the Author):

Summary: The authors used serial questionnaire data from the long-COVID in Scotland Study (Long-CISS) to examine long-COVID symptoms in a general population cohort of adults (>16 years) with laboratory-confirmed SARS-CoV-2 compared with a matched (age, sex and index of deprivation) group of adults without infection. The study has novel data that is often not available (self-reported symptoms, negative control group, follow-up for more than 6 months) in long COVID studies and will be a nice addition to the literature. However, I have concerns with some of the methods and results reporting. My major comments:

- 1) The authors do not acknowledge any limitations of their analysis or design. The authors overstate the non-selective nature, given participants have to respond and complete a minimum of two questionnaires. Additionally, in any observational study, confounding is a concern. And I am concerned about the table 2 fallacy and bias for some of the presented results.
- 2) Is there bias due to using a comparison group that 'never' tests positive versus, say, comparison with a group that tests negative at the same time as a positive test? This is difficult to assess without a better description of how the authors selected negative testers (please see methods, point 2 comment).
- 3) Suggest reconceptualizing recovery as one outcome. You identify the same set of predictors for improvement and deterioration (older age and depression), which is confusing and perhaps unhelpful for planning or prioritizing health/social care.
- 4) Why did you choose to show 5 symptoms in the adjusted models? Did you select significant or interesting findings from table 4 to model in table 5? For contextualizing the many long COVID studies without a SARS-CoV-2 negative comparison group, presenting null effects would be helpful, as would highlighting if the SARS negative group had greater odds of reporting symptoms than SARS positive group.

Minor points

Methods:

- 1) Study period is not specified. SARS-CoV-2 PCR results from April 2020 to what?
- 2) Negative testers.
 - a. Why use most recent negative test date? Does this bias results such that your negative testers are selected more recently than your positive testers? And if so, how might that impact symptom reporting?
 - b. Are you selecting people without a positive test in the 18 months of follow-up or without a negative test as of the end of your study period? If the latter, are those people who are never testing positive, at this point in the pandemic, fundamentally different, from people who have SARS-CoV-2 infection?
 - c. How do rapid tests bias results? Do you expect that some of the negative testers selected later in the study have actually had SARS-CoV-2 that wasn't reported as a PCR test? Should this limitation be acknowledged?
 - d. How does your matching algorithm change if someone is later found to test positive? Did you consider matching on index date, which may be important for seasonal reporting of symptoms?
- 3) Outcome definitions.
 - a. The quality of life, symptoms and recovery status are listed as outcomes but undefined and unclear. E.g., are these validated measures? Are you assessing presence or change at certain months?
 - b. Recovery appears to be two binary outcomes. But the details are unclear. What happens to the group that is consistently bad or consistently good – are they in the denominator? Having no improvement (consistently bad) seems different than consistently good and may be better classified as deterioration / no recovery.
- 4) Models. Suggest you rephrase as covariates (line 222). Confounders implies you have an exposure in mind and you appear to be looking at all these factors as they relate to recovery.

Results:

1. A study flow diagram may make it easier to understand how many people are included overall and for each research question. Also suggest describing who is in the analysis.
2. Line 258-261. Reported numbers don't match table 1. E.g., 55% report full recovery at 6 and 12 months? Suggest double checking that your labels in text narrative match labels in tables.
3. Lines 263-269. Difficult to follow. Suggest adding to table 1. Are you pulling pulling numbers across constant, deteriorated, and improved categories?

4. Line 271. Is the p-value for comparison among just the SARS-CoV-2 infected? It also looks like there are more women. Why the focus on depression effect vs other significant differences?
5. Table 4. Lines 284-286, unclear. What symptoms are you are comparing?
6. Lines 288-290. Sentence is difficult to follow. Suggest you use the same labels between Table 4 and 5 (change in taste is the same as altered taste)? Highlighting the prevalence would be beneficial (preferably adding these details to Table 5).
7. Your primary aim was to compare symptoms for positive versus negative SARS-CoV-2. The table 2 fallacy may be a problem for Table 5 and e.g., line 299 or 308. Suggest re-framing these finding of your secondary risk factors in your methods and results. (see e.g., Westreich, The Table 2 Fallacy: Presenting and Interpreting Confounder and Modifier Coefficients),
8. Table 5. Title and outcome are unclear. Your reference is the exposure. But your title suggests you've made the outcome the reference. If the outcome is change from 6-12 months, this should be clarified. Please add number of people in the analysis.

Reviewer #3 (Remarks to the Author):

General comments

The authors have conducted a follow on analysis of a cohort exploring the illness trajectory of long covid. The manuscript is well written and data analysis sound. However, I do not think this present manuscript provides a substantial advancement of our knowledge of the condition beyond that provided by the previous article by the authors.

Specific comments

- The authors compared the study population with a control group that 'never' had covid-19. I think they need to be more tentative here – having a negative PCR test does not mean a person never had covid. Their bodies could have cleared the virus by the time of testing, and it is quite possible that some symptoms may persist even in the absence of the virus. Please rephrase perhaps using a phrase such as 'PCR-negative' or you could state earlier and clearly in the manuscript that you would refer to the PCR-negative controls as 'never infected' even though you are aware this may not always be the case.
- It might be helpful for the reader if the authors used subheadings to signpost each of the symptoms under the 'changes in symptoms' section of the results (pages 11 & 12). Depending on journal style these could be written in italics.
- The authors provided the statistical results and p-value for prevalence of confusion at the 6-month period (page 11). Is there a reason for this as they did not provide for any other statistically significant result in the section. If word count permits, consider including all statistically significant results throughout the manuscript.
- The authors mentioned in the results and method section the calculation of median EQ 5D scores. Reading through the results I am convinced they were referring to EQ 5D VAS scores. Please indicate throughout the manuscript that you are referring to VAS scores as readers might wrongly assume you are referring to index scores.
- Given the potential association between depression and quality of life, did the authors conduct any analysis to investigate in this study?
- The authors focused mostly on symptoms with significant changes in the result section. However, some of these are not necessarily the most troublesome for patients. While there were no significant changes for the PCR positive participants in symptoms like fatigue, chest pain, breathlessness (Table 4), these are findings that should get mentioned perhaps in the discussion section as they have implications for individuals with long covid.
- The authors stated on page 13 that 'late onset cough and hearing problems were new findings'. This is not entirely correct as their previous analysis from the same study cohort showed 'increased reporting of a dry or productive cough between 6 and 18 months' which remained significant for cough with phlegm when participants at 12 and 18 months were compared (Supp Table 3) (Hastie et al 2022, Nat Comms).

Reviewer #1

(Remarks to the Author):

This is a valuable study to help understand how adults with on-going symptoms following SARS-CoV-2 infection and one of the first to look at symptoms at 12 and 18 months in a community based population. However, there are a number of detailed comments below that could strengthen the paper. One overall comment, is that it is difficult to track how participants are followed over time and who is and isn't included in the denominator (and numerator) for each estimate. A clear flow diagram on documenting which participants are including in the analyses is needed.

Thank you for your comments. A flow diagram has been added.

In terms of presentation of the results, recommend reorganizing the results to strength the messages of the paper.

Recommendations:

Table 1- comparing characteristics of participants at each time point (6 month, 12 month, 18 months) by infected and not infected status. This will support whether there was differential drop out of participants by infectious status.

Under the data sharing agreement we have with the data custodian we are not provided with individual level data on e.g. infection status or response rate. The only aggregated data that Public Health Scotland has been billing to share is the percentage of never infected who subsequently become infected and the overall percentage who respond at each stage of follow-up. This is not sufficient to stratify response rates at different stages of follow-up by infection status because we do not know if those who change infection status responded or not. However, in response to the referee's comment we have added differential attrition rates as a potential limitation to the discussion section.

Table 2- look at change in report of symptoms from 6 months at 12 months, and then at 18 months- also by infection status.

The cohort study was ambi-directional (e.g. some were recruited after 6 months follow-up), not everyone has reached 18 months follow-up and people were able to complete questionnaires or not at each stage (e.g. they may have completed 6 and 18 months but not 12 months). Therefore, the people who had questionnaires at all three follow-up points (6, 12 and 18 months) is only a sub-group of the people who completed questionnaires at 6 and 12 months and of those who completed questionnaires at 6 and 18 months. Restricting the analysis to people who completed the questionnaire at all three points would severely reduce power, not only in comparison to 6 vs 12 months but also in comparison to 6 vs 18 months. Hence, we presented the table as two comparisons – 6 vs 12 and 6 vs 18 – in order to maximise statistical power.

Figure looking at persistent, recovered, worsening symptoms from 6 to 12 months, and then from 6 to 18 months, by infectious status. Also consider looking at symptoms from 12 to 18 months.

We do not have self-reported recovery status for the comparison group. However as requested below, a graph based on 18 month follow-up for population (denominator) with a stacked bar with % no symptoms at 18 months, % new symptoms from 6 months, % persistent symptoms from 6 months has

been added as a Supplementary Figure, with a similar figure for 12 months. We have insufficient statistical power to compare 12 and 18 months.

Please see comments for how to approach the multivariate analysis.

Specific comments:

Summary of article:

- The authors state that the not infected group was matched to the infected group. If so, what factors were they matched on? If not, then recommend revising this wording.

This is already included under methods/study design and participants but have also added it to the summary in response to the referee's comment. The comparison group was matched on sex, age, Scottish Index of Multiple Deprivation quintile, and time period of PCR test.

- Is there any information on symptoms at time of infection or testing?

This information was reported in detail in our previous publication (below). In this paper it was only used to exclude people with asymptomatic infections as it was not relevant to the aim.

Hastie CE, Lowe DJ, McAuley A, Winter AJ, Mills NL, Black C, Scott JT, O'Donnell CA, Blane DN, Browne S, Ibbotson TR, Pell JP. Outcomes among confirmed cases and a matched comparison group in the Long-COVID in Scotland Study. Nat Commun 2022;13(1):5663 doi: 10.1038/a41467-022033415-5.

- Please explain how “fully, partially and not recovered” compares to “persistent and recovered”. Based on the methods it looks like fully and partially recovered is based on self assessment, but then persistent symptoms are based on reporting symptoms at 2 time points. Is that correct? This applies to reporting in the main results also.
- Please see the comments related to prevalence of persistent symptoms in the main results and apply here also.

There are two completely separate outcomes. Recovery status was self-reported by people previously infected. Individual symptoms were self-reported by both groups. Recovery status at each follow-up point was self-categorised as fully, partially or not recovered. Serial self-reports in the same (previously infected) individual were used to categorise an individual's recovery status trajectory as either: improvement in recovery status (change from no recovery to partial/full recovery, or from partial recovery to full recovery), deterioration in recovery status (change from full recovery to partial/no recovery or from partial recovery to no recovery) or persistent incomplete/no recovery (partial recovery to partial recovery or no recovery to no recovery).

Serial self-reporting of symptoms by the same individual was used to identify people in whom a symptom was persistent (reported at time 1 and time 2), resolved (reported at time 1 but not reported at time 2) or late onset (not reported at time 1 but reported at time 2). This is explained in detail in the methods section. However, the submission guidelines state that the summary paragraph should be around 200 words. The current word count is 215 therefore we cannot also explain it in this detail in the summary.

Methods:

- What was the sample size and response rate?

The sample size was 23,973 (12,947 symptomatic infected and 11,026 never infected) for the primary analysis of 6 and 12 month follow-up. This has been clarified in the text. Overall, the response rate was 9%. However, it is not possible to calculate the response rate separately for the infected and comparison groups for the reasons described above.

- Can you provide the survey instrument in an on-line supplement?

The questionnaire has been added as a supplement.

- Were pre-existing health conditions based on pre-existing conditions prior to SARS-CoV-2 infection or prior to 6 month survey?

Pre-existing conditions were prior to the index PCR test (i.e. the PCR test used to categorise individuals into the infected or comparison group).

- Was linkage to ambulatory care also included?

No. In Scotland ambulatory care is recorded in a separate database – SMR00. Unlike hospital admissions (SMR01), no disease codes are recorded on SMR00 records.

- What is the completion rate of linkages?

Linkage is based on exact matching using the CHI (Community Health Index). Everyone born in Scotland or born elsewhere and now resident in Scotland is allocated a CHI. The only people who would have been unable to be linked would be people living in other countries who had a positive PCR test whilst visiting Scotland. Obviously, the number of visitors will have been less during the covid-19 pandemic due to travel restrictions. Also, their exclusion does not impact on the aim of the study which was to understand the natural history of SARS-CoV-2 infection among Scottish residents.

- If a participant who tested negative later tested positive how was this addressed by the study?

These people are reallocated to the infected group on the date of the positive PCR and their follow-up in this new group starts on this date. So, for example someone could provide information on 6 month follow-up in the comparison group and then 6 month follow up post infection. In reality, these people are less likely to contribute data to this study (compared to our previous study) due to the need to have serial outcome measurements in the same individual in the same group.

- Were quality of life scores asked for both those infected and not infected?

Yes.

- Were the quality of life scores reported in the results?

Median EQ-5D scores are reported with interquartile ranges for both symptomatic infected and uninfected individuals at 6 and 12 month follow-up. Please see the last paragraph of the results.

- Did you consider analysis where you looked at symptoms at all three time points- 6, 12 and 18 months?

This was considered but was not possible due to the impact on statistical power as explained above.

- In the paragraph describing how the change in prevalence was calculated, it is not clear what was done. Suggest revising to state clearly- e.g. "change in prevalence of symptoms from 6 to 12 months was compared among those infected and never infected.... Next change in prevalence of symptoms between 6 and 18 months was compared.... Finally, trend in symptoms from 6 to 12 to 18 months was compared..."

We have amended the text as suggested, with the exception of the final sentence because we did not examine trends from 6 to 12 to 18 months.

- Why was poisson regression chosen for this analysis? What other factors were included in the model?

Poisson regression was used only for the analysis of EQ-5D as an outcome because responses are count data. All multivariate analyses were adjusted for age, sex, deprivation quintile, ethnic group, individual and total number of long-term conditions, vaccination status, and dominant variant.

- Recommend that models be stratified by infection status. This will allow a comparison of what factors are related to symptoms at the different time points among those not infected and those infected separately.

The aim of the study was to determine the natural history of long-covid. The purpose of the comparison group was to control for symptoms that would have occurred anyway even in the absence of infection. Determining the factors associated with symptoms in the infected group only would not take account of the fact that most of the symptoms post covid are common and generic and do not signify long-covid. And determining the factors associated with symptoms in the never infected would not contribute any useful insights.

Results:

- Strongly recommend adding a flow diagram for the participants, both infected and not infected, including who remains in the study at 12 and 18 months.

Thank you for the suggestion. A flow diagram has been added.

- Were there any differences in loss to follow-up by infection status of demographic characteristics? If there were differences, how was this addressed in the analysis and how did this likely impact the results?
As explained above we do not have access to this information. This has been added as a study limitation.

- This manuscript would benefit from a graph that is based on 18 month follow-up for population (denominator) with a stacked bar with % no symptoms at 18 months, % new symptoms from 6 months, % persistent symptoms from 6 months; could do a similar figure for 12 months. Figures could be shown for symptoms overall and by type and could also produce figures by demographics.

This has been added as a Supplementary Figure.

- Recommend a table that reports the characteristics of participants SARS-CoV-2 + at 6 months, then looks at who also completed the survey at 12 m and then 18 m to look at loss of follow-up and how this may have changed across characteristics over time. If there are changes, this should be address in analysis or mentioned in limitations.

As explained above, due to the ambidirectional nature of the cohort, not all participants started with 6 months follow-up, some will have started at 12 months follow-up and very few participants provide data at 6, 12 and 18 months. Therefore, it is impossible to report changes in characteristics over the follow-up period unless based on a small sub-group which is unlikely to be representative.

- Table 1. Are the report of resolved symptoms in table 1 based on response that specific survey question and then compared to whether or not participants reported symptoms at both time points? How does this compared to the not infected group.

Table 1 is based on changes in self-reported recovery status (full/partial/no recovery) between the two timepoints. It is not based on self-reported symptoms and individuals in the never infected comparison group were not asked to report their recovery status as they did not need to recover.

- Table 2. Recommend moving this to the first table and including report of symptoms in the never infected group have survey questions at multiple time points.

Table 2 provides the characteristics of the whole cohort and therefore includes the never infected comparison group which was not included in Table 1. For this reason it would be impossible to merge Table 2 into Table 1. Table 2 reports the characteristics at baseline and therefore does not report outcomes such as symptoms at 6, 12 and 18 months. Outcomes are reported in the subsequent tables. Table 1 needs to precede Table 2. Because it explains how the 2nd, 3rd, and 4th columns in Table 2 are derived.

- Table 3. Please clarify how the sample in table 3 aligns with what is reported in table 1 and 2.

Table 1 reports on the 12,947 people with previous infection who provided 6 and 12 month follow-up data as well as the 4,196 people with previous infection who provided 6 and 18 month follow-up data. Table 2 reports on the 11,026 people in the never infected comparison group who provided 6 and 12 month follow-up data and who were not included in Table 1. Table 2 also reports on the 12,947 previously infected people who provided 6 and 12 month follow-up data (left hand column in Table 1): 9,839+1,497+1,611=12,914.

Table 3 is again based on previously infected individuals who provided 6 and 12 month follow-up and, in terms of how four columns in Table 3 relate to the columns in Tables 1 and 2:

Improved referent to no change: 12,947 previously infected with follow-up at 6 and 12 months excluding 6,407 people who were already fully recovered at 6 months follow-up (so could not improve) and excluding people whose recovery status deteriorated (458 after deducting those already excluded) = 6,082

Improved referent to no change or deterioration: 12,947 previously infected with follow-up at 6 and 12 months excluding 6,407 people who were already fully recovered at 6 months follow-up (so could not improve) = 6,540. Other people who deteriorated are included in the referent category so not excluded.

Deterioration referent to no change: 12,947 previously infected with follow-up at 6 and 12 months excluding 891 people who reported no recovery at 6 months (so could not deteriorate) and excluding people whose recovery status improved (1,179 after deducting those already excluded) = 10,877

Deterioration referent to no change or improvement: 12,947 previously infected with follow-up at 6 and 12 months excluding 891 people who reported no recovery at 6 months (so could not deteriorate) = 12,056. Other people who improved are included in the referent category so are not excluded.

A footnote has been added to the table to clarify this.

- The prevalence of symptoms in the never infected group is 53% compared to the infected group this is a difference of 18% which is what has been reported in multiple other studies. given the strength of this study is the never infected group, this comparison should be reported in the results.

The prevalence of at least one symptom is reported in the last row in Table 4. It is reported by infected vs comparison group and at different stages of follow-up

- Did you also look at report of 2 or more symptoms?

No. We applied the standard definition of long-covid based on at last one persistent or new symptom.

- Table 5. This model doesn't make sense to me- what exactly is the outcome? symptoms at 12 months among participants reporting the symptom at 6 months? if so then one would say, among participants who had altered taste at 6 months, those who had infection were less likely to have altered taste at 12 months compared to those not infected (after adjusting for xxxx) this does make sense with table 4- but looking at table 4 it seems like it may be driven by small numbers in the never infected group

The aim of the study is to describe the natural history (i.e. trajectory) of long-covid so this table is analysing changes in symptoms. The Table models the likelihood of having the symptom at e.g. 12 months adjusted for whether it was present at 6 months and adjusted for confounders. Hence resolution or late onset of symptoms.

- why not stratify by infection status -then report out factors associated with persisted symptoms if infected, factors associated with persistent symptoms in general?

That would negate the aim and strengths of the study. The reason for having a negative test comparison group is because all of the symptoms associated with long-covid (other than altered taste and smell) are very non-specific symptoms that are common in the general population. Therefore, the model needs to take account of the likelihood of these symptoms occurring in the comparison group as well as differences between the two groups that might lead to confounding.

- Supplemental Table 4 is recovery defined at 12 months? I am unclear based on this table. is change in taste at 6 months change from baseline?

Based on the referee's comment, we believe the labelling of the rows may be misleading. The figures are the prevalence of e.g. altered taste at 6 months and at 12 months. Therefore, you can see if the crude prevalence changed between these two time points. To avoid confusion we have changed the labels to "Altered taste – 6 months – 12 months" etc.

Discussion:

- Paragraph 1 in the discussion describing the trajectories I would frame this as each symptom as a different trajectory and that because of this it is important to look at individual symptoms and not group together.

The discussion text has been amended accordingly.

- In reporting out the prevalence of symptoms at 12 and 18 months, it is often not clear to who is included in the denominator is for the estimate. For example, is it among people who had a symptom at 6 months, meaning that 70% of symptoms persisted or is it among all people who were sars-cov-2 positive? Please clarify this, both in the results and discussion.

Percentages were calculated with the denominator as the whole sample at each timepoint within each exposure group.

- It is also important to compare the estimated prevalence among the infected group with the not infected group. The prevalence of symptoms in the not infected group is also high.

We have added a comment on the prevalence in the not infected comparison group.

- What does the difference over time in symptoms look like in the not infected group? This is an important point to include in the discussion and to help put the results into context.

This information is already contained in Table 4.

Reviewer #2

(Remarks to the Author):

Summary: The authors used serial questionnaire data from the long-COVID in Scotland Study (Long-CISS) to examine long-COVID symptoms in a general population cohort of adults (>16 years) with laboratory-confirmed SARS-CoV-2 compared with a matched (age, sex and index of deprivation) group of adults without infection. The study has novel data that is often not available (self-reported symptoms, negative control group, follow-up for more than 6 months) in long COVID studies and will be a nice addition to the literature. However, I have concerns with some of the methods and results reporting. My major comments:

- 1) The authors do not acknowledge any limitations of their analysis or design. The authors overstate the non-selective nature, given participants have to respond and complete a minimum of two questionnaires. Additionally, in any observational study, confounding is a concern. And I am concerned about the table 2 fallacy and bias for some of the presented results.

We have expanded the limitations section in the discussion to address this.

- 2) Is there bias due to using a comparison group that 'never' tests positive versus, say, comparison with a group that tests negative at the same time as a positive test? This is difficult to assess without a better description of how the authors selected negative testers (please see methods, point 2 comment).

The comparison group invitees were matched to the infected group on time period of the index test. This has been added to the summary and methods.

- 3) Suggest reconceptualizing recovery as one outcome. You identify the same set of predictors for improvement and deterioration (older age and depression), which is confusing and perhaps unhelpful for planning or prioritizing health/social care.

Recovery status is one outcome but, in relation to the natural history analysis, it has three values: constant, deterioration and improvement. Our study design, analysis plan, and interpretation were developed in discussion with the University of Glasgow, College of Medical, Veterinary and Life Sciences COVID-19 PPIE (Patient and Public Involvement and Engagement) group. Patient representatives from this group felt it important to have this level of granular detail in the outcome. For example, they did not view deterioration as equivalent to not improving. From a methodological perspective, there is no reason to assume that a factor that increases the likelihood of improvement would necessarily reduce the likelihood of deterioration, since remaining in the same state is also a possibility.

- 4) Why did you choose to show 5 symptoms in the adjusted models? Did you select significant or interesting findings from table 4 to model in table 5? For contextualizing the many long COVID studies without a SARS-CoV-2 negative comparison group, presenting null effects would be helpful, as would highlighting if the SARS negative group had greater odds of reporting symptoms than SARS positive group.

We selected these covariates because they were statistically significant in Table 4. They are being included in the models as potential confounders and therefore by definition need to be associated with both the exposure and outcome.

Minor points

Methods:

- 1) Study period is not specified. SARS-CoV-2 PCR results from April 2020 to what?

The test date range for participants in the final sample has been added to the results.

- 2) Negative testers.

- a. Why use most recent negative test date? Does this bias results such that your negative testers are selected more recently than your positive testers? And if so, how might that impact symptom reporting?

The comparison group invitees were matched to the infected group on time period of the index test so this should not have introduced time bias. We have added this information to the revised manuscript.

b. Are you selecting people without a positive test in the 18 months of follow-up or without a negative test as of the end of your study period? If the latter, are those people who are never testing positive, at this point in the pandemic, fundamentally different, from people who have SARS-CoV-2 infection?

The study was ambidirectional, recruiting people both retrospectively and prospectively. Therefore, people will have completed their last questionnaire at different points in the pandemic. However, it is true that attrition of the comparison group (due to a positive PCR and change in group) increased from 6 to 12 to 18 months follow-up. We have added to the limitations section of the discussion that this may have introduced systematic differences.

c. How do rapid tests bias results? Do you expect that some of the negative testers selected later in the study have actually had SARS-CoV-2 that wasn't reported as a PCR test? Should this limitation be acknowledged?

We did not have access to the results of lateral flow tests and testing practice changed over time. Therefore, we excluded 53,530 people who had no record of a positive PCR test but who reported having had covid-19. We have amended the revised manuscript to make this clear.

d. How does your matching algorithm change if someone is later found to test positive? Did you consider matching on index date, which may be important for seasonal reporting of symptoms?

If someone in the negative PCR comparison group later tests positive, they are re-allocated to the infected group with new t=0 on the positive test date. We started the study with 3:1 matching to allow for change in classification of the negative comparison group and ensure that this did not adversely affect statistical power. As mentioned above we did match on index date. We are sorry that this was not clear.

3) Outcome definitions.

a. The quality of life, symptoms and recovery status are listed as outcomes but undefined and unclear. E.g., are these validated measures? Are you assessing presence or change at certain months?

We report the three outcomes at 6, 12 and 18 months and the changes in them between 6 and 12 months and between 6 and 18 months. The questionnaire has been added as a supplement. Quality of life is assessed using a validated tool; the EQ-5D visual analogue scale. Symptoms and recovery status questions were harmonised with the ISARIC questionnaire. This reference has been added to the text: Sigfrid, L. et al. What is the recovery rate and risk of long-term consequences following a diagnosis of COVID-19? - A harmonised, global longitudinal observational study protocol. BMJ Open 11, e043887 (2021)

b. Recovery appears to be two binary outcomes. But the details are unclear. What happens to the group that is consistently bad or consistently good – are they in the denominator? Having no improvement (consistently bad) seems different than consistently good and may be better classified as deterioration / no recovery.

Recovery status is an ordinal variable: fully recovered, partially recovered and not recovery. Change in recovery status is an ordinal variable: constant, deterioration and improvement. As explained in the

responses to referee 1, people who were already fully recovered were not included in the model for improvement since they could improve. Similarly, people who reported no recovery were not included in the model for deterioration. A footnote has been added to the table to clarify this.

- 5) Models. Suggest you rephrase as covariates (line 222). Confounders implies you have an exposure in mind and you appear to be looking at all these factors as they relate to recovery.

The wording has been amended.

Results:

1. A study flow diagram may make it easier to understand how many people are included overall and for each research question. Also suggest describing who is in the analysis.

Thank you for the suggestion. A flow diagram has been added.

2. Line 258-261. Reported numbers don't match table 1. E.g., 55% report full recovery at 6 and 12 months? Suggest double checking that your labels in text narrative match labels in tables.

41% refers to the percentage who report full recovery at 6 and 12 months out of the whole sample (i.e. 5,368/12,947), rather than out of only those with constant recovery (which is 5,368/9,839 = 55%).

3. Lines 263-269. Difficult to follow. Suggest adding to table 1. Are you pulling pulling numbers across constant, deteriorated, and improved categories?

Yes, these numbers are derived from addition of the relevant rows across Table 1.

4. Line 271. Is the p-value for comparison among just the SARS-CoV-2 infected? It also looks like there are more women. Why the focus on depression effect vs other significant differences?

Yes, the p-value for comparison is among the infected group. The association with depression is consistent between 6 and 12 month, and 6 and 18 month, follow-up. We have added the association with socioeconomic deprivation quintile which is also consistent.

5. Table 4. Lines 284-286, unclear. What symptoms are you are comparing?

We are comparing the proportion of people with one or more symptom at 6 and 12 months, and 6 and 18 months, separately for symptomatic infected and never infected groups (the bottom line of Table 4). These prevalence estimates have been added to the text.

6. Lines 288-290. Sentence is difficult to follow. Suggest you use the same labels between Table 4 and 5 (change in taste is the same as altered taste)? Highlighting the prevalence would be beneficial (preferably adding these details to Table 5).

The text has been amended so the labels are the same throughout.

7. Your primary aim was to compare symptoms for positive versus negative SARS-CoV-2. The table 2 fallacy may be a problem for Table 5 and e.g., line 299 or 308. Suggest re-framing these finding of your secondary risk factors in your methods and results. (see e.g., Westreich, The Table 2 Fallacy: Presenting and Interpreting Confounder and Modifier Coefficients),

Thank you for providing this reference. We have added a note in the text stating that these are secondary effect estimates.

7. Table 5. Title and outcome are unclear. Your reference is the exposure. But your title suggests you've made the outcome the reference. If the outcome is change from 6-12 months, this should be clarified. Please add number of people in the analysis.

We have amended the Table title to clarify. The outcome is whether the symptom is present at 12 months and it is adjusted for whether the same symptom was present at 6 months.

Reviewer #3

(Remarks to the Author):

General comments

The authors have conducted a follow on analysis of a cohort exploring the illness trajectory of long covid. The manuscript is well written and data analysis sound. However, I do not think this present manuscript provides a substantial advancement of our knowledge of the condition beyond that provided by the previous article by the authors.

We disagree. The previous paper described the epidemiology of long-covid (frequency, distribution and determinants). This paper uses serial measurements in the same participants (unlike the previous study) to describe the natural history (trajectory) of long-covid; i.e. once present does long-covid remain static, resolve or progress and in whom.

Specific comments

- The authors compared the study population with a control group that 'never' had covid-19. I think they need to be more tentative here – having a negative PCR test does not mean a person never had covid. Their bodies could have cleared the virus by the time of testing, and it is quite possible that some symptoms may persist even in the absence of the virus. Please rephrase perhaps using a phrase such as 'PCR-negative' or you could state earlier and clearly in the manuscript that you would refer to the PCR-negative controls as 'never infected' even though you are aware this may not always be the case.

The comparison group never had a positive PCR test and reported never having had COVID-19. Therefore, people who had a positive lateral flow test, clinically diagnosed COVID-19 or self-diagnosed COVID-19 infection were not included in this group. We agree that the group may still include a small number of people with e.g. asymptomatic infections but this is the best definition achievable without regular antibody testing of the whole population. We defined never infected in the methods section.

- It might be helpful for the reader if the authors used subheadings to signpost each of the symptoms under the 'changes in symptoms' section of the results (pages 11 & 12). Depending on journal style these could be written in italics.

These subheadings have been added to the text.

- The authors provided the statistical results and p-value for prevalence of confusion at the 6-month period (page 11). Is there a reason for this as they did not provide for any other statistically significant result in the section. If word count permits, consider including all statistically significant results throughout the manuscript.

The association of long-covid with depression is of particular interest to academics, the public and media. Hence, we provided this additional information for this specific issue.

- The authors mentioned in the results and method section the calculation of median EQ 5D scores. Reading through the results I am convinced they were referring to EQ 5D VAS scores. Please indicate throughout the manuscript that you are referring to VAS scores as readers might wrongly assume you are referring to index scores.

Yes, you are correct we are referring to the EQ-5D visual analogue scale. We have clarified the text accordingly.

- Given the potential association between depression and quality of life, did the authors conduct any analysis to investigate in this study?

Not in the current study because the primary focus of the study was the natural history of long-covid rather than the impact of long-covid.

- The authors focused mostly on symptoms with significant changes in the result section. However, some of these are not necessarily the most troublesome for patients. While there were no significant changes for the PCR positive participants in symptoms like fatigue, chest pain, breathlessness (Table 4), these are findings that should get mentioned perhaps in the discussion section as they have implications for individuals with long covid.

This is an important point and highlights the limitations of the current definitions of long-covid. Definitions are based on e.g. having one or more persistent symptom irrespective of which symptom is present. In reality, the 'impact' of long-covid is likely to vary; in part reflecting the number of symptom, their severity and which symptoms are present. We did not measure the severity nor impact of the symptoms so cannot address this question directly but, as requested, have added a line to the discussion to state that the impact of improvements or deterioration in symptoms is likely to vary depending on which symptoms occur and their original severity.

- The authors stated on page 13 that 'late onset cough and hearing problems were new findings'. This is not entirely correct as their previous analysis from the same study cohort showed 'increased reporting of a dry or productive cough between 6 and 18 months' which remained significant for cough with phlegm when participants at 12 and 18 months were compared (Supp Table 3) (Hastie et al 2022, Nat Comms).

We have clarified the text and referred to our previous findings.

REVIEWERS' COMMENTS

Reviewer #1 (Remarks to the Author):

Thank you for addressing the comments. The paper is much improved. I have no additional comments.